# Modelling the Application of Telemedicine in Emergency Care

**Gyoergy (George) L. Ferenczi [1],\* and Áron Perényi [2]**

1 Engineering and Mathematical Sciences, School of Molecular Sciences, La Trobe University, Melbourne, VIC 3086, Australia
2 Department of Business Technology and Entrepreneurship, School of Business, Law and Entrepreneurship, Swinburne University of Technology, Hawthorn, VIC 3122, Australia; aperenyi@swin.edu.au
\* Correspondence: georgeferenczi@yahoo.com.au; Tel.: +61-415327620

**Abstract:** Emergency services are under pressure worldwide. Ambulance services in Victoria in Australia are particularly overloaded and the quality of service is suffering in comparison to other health services in Australia. An abundance of articles addresses this issue both in academic and industry outlets, and the proposed solutions usually advise upgrades and better use of available resources. We believe that telemedicine could be part of the solution. Patients can be quickly assessed and monitored by advanced medical sensors, connected by straightforward means including a direct video link, to the hospital. Pre-assessment of conditions can be sent ahead to the emergency department, where specialists and physicians can select priorities and prepare for urgent interventions. An increasing number of patients with mental health, drug or alcohol-related issues can be transported elsewhere, thus reducing the load of emergency departments. We have methodically analysed Victorian ambulance statistics and we have identified appropriate telemedical technologies to be used in appropriate settings. We applied telemedical technology models in our work, to demonstrate the potential improvements in outcomes, including patient lives saved.

**Keywords:** telemedicine; emergency services; tele-cardiology; care efficiency

## 1. Introduction

### 1.1. Background and Context

Emergency medical response services in Victoria, Australia are provided by Ambulance Victoria, on a territory of approximately 220,000 square kilometres, for almost 6 million people [1]. While the number of emergency service medical responses is growing at an annual rate above 10%, the sector is struggling with numerous issues, including staffing and service outcomes.

Based on an observational study [2] conducted of consecutive cases attended by Ambulance Victoria in Melbourne, Australia from 2008 to 2015, a substantial growth in demand was identified. Incidence rates calculated, using time series regression analyses of ambulance demand based on a total of 2,443,952 consecutive cases, showed that demand grew by 29.2% over the 8-year period. Incidence rates increased significantly over time for patients aged over 60 years, but to a less extent for patients aged under 60 years. An overall annual growth of 1.4% in demand was observed over the examined time period, with growth of incident rates in patients with a history of mental health issues (5.8% growth per annum), alcohol/drug abuse (6.1% growth per annum) or multiple compounding health issues (4.5% growth per annum). Case numbers involving patients with socio-economic or educational disadvantage, younger age groups, or no preexisting health conditions also grew faster than 1.4%. The number of cases requiring hospital transport increased by 1.2% annually, while the number of cases not requiring medical intervention from paramedics increased by 6.7% annually. The increase in service demand in cases in these specific areas exceeded the population growth in Victoria between 2008 and 2015 (1.3 to 2.4% annually [3]). As the distribution of patient categories demonstrated, emergency

ambulances were increasingly utilised to transport patients who did not need medical intervention from paramedics. This situation outlined above has only evolved in a less favourable direction, with emergency medical service demand continuously growing [4], while capacities and service levels waver [5].

In Victoria, the Emergency Services Telecommunication Authority (ESTA) is responsible for handling the incoming emergency calls [6]. These calls are responded to by means of a computer-aided dispatch (CAD) system [7]. Unfortunately, however, these systems fail to meet the needs of a modern and responsive emergency system [8]. There have been attempts in Australia to introduce other technological solutions (such as an emergency service smartphone app developed by Fire and Rescue NSW [2]), to utilise benefits of technology (such as GPS location of the caller). The need for such solutions is justified by horrible examples of allegedly mishandled cases and fatalities as outcomes (i.e., the cases of David Iredale in 2006 [9] and Joanne Wicking in 2011 [10]).

*1.2. The Problem*

A triage system is implemented at Victorian emergency stations and hospitals [11]. The triage begins when the patient actually gets into the emergency department. With the significant increase in emergency medical service demand in Victoria [12], in many cases, the ambulances are standing in line in front of the hospital with the patients in them [13,14]. According to the triage procedure, the hospital staff has very little knowledge of the actual status of the patients waiting in the ambulances (see Figure 1). However, offloading them more quickly into the emergency departments does not fix the problem about additional risks to patient health [15]. Due to the overload, patients may end up waiting a long time before they are attended to by emergency department staff [16], and paramedics may need to step in and provide services prior to the emergency department to alleviate the pressure [17], or an improvement of the workflow is needed to increase efficiency by building on existing resources [16]. Prioritisation of certain ambulant patients is low [18], and something as simple as ambulance-based telephone triage—connecting patients, paramedics and emergency departments [19]—is known to effectively improve emergency referral outcomes.

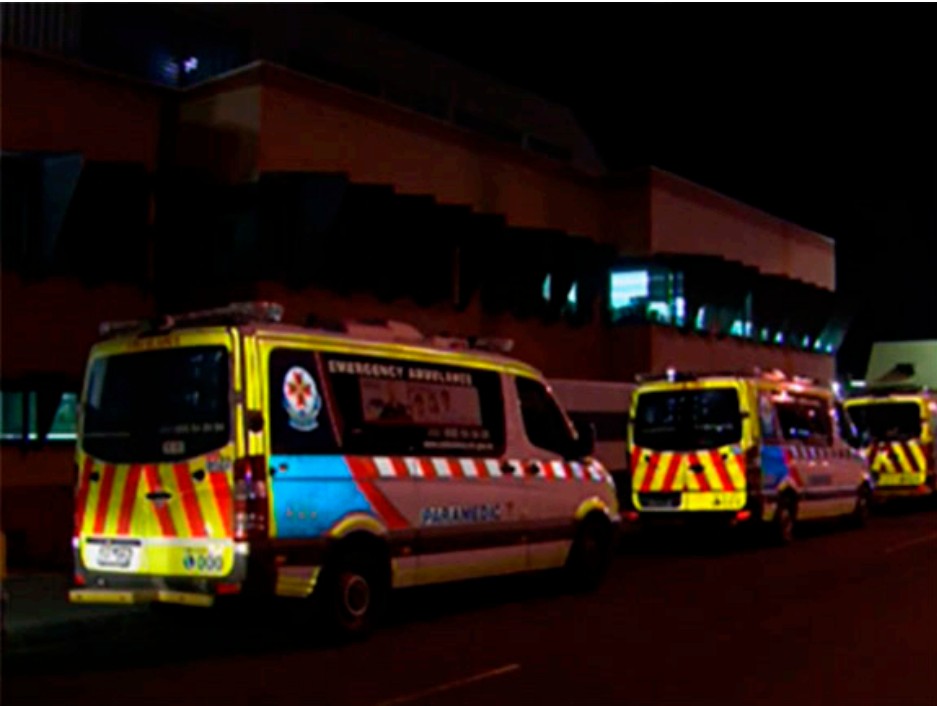

**Figure 1.** Ambulances waiting in line at a hospital in Victoria.

Emergency service communications are state of the art. Dispatch operators use a range of technologies including digital and analogue radio, telephone, and (rarely) TCP/IP-based communications and geographical positioning systems. Ambulances are often equipped with data terminals that enable them to view information, read messages sent by call-takers and dispatchers, and be notified of updates. Digital radio and mobile data terminals are generally unavailable outside greater Melbourne [20].

*1.3. The Process*

The current practice is to classify the incoming patients by a process called triage. The patients left at the end of the triage line (with a perceived minor health issue only) are often waiting for several hours in the emergency waiting area, after which they are simply told to go home. To support this process, the hospital employs a strong security workforce who often intervene when the patients become impatient and upset after waiting and dismissal. A more civilised means of treatment would be to set up telemedical mental health therapist support and/or general practitioner consultancy for these leftover patients. The therapists could be anywhere, even overseas; for this, the necessary legislation would need to be changed. Any solution would be better than the currently applied usual practice, which is very poor.

*1.4. The Proposed Solution*

Based on available statistics, IT simulations, our former experience and modelling estimations, we present a telemedical system that can deliver significant improvements in quality of care and the cost-to-efficiency ratio.

**2. Materials and Methods**

*2.1. Modelling*

We studied a range of medical and healthcare models [21–25] as well as telemedical systems prior to engaging in this simulation. We created both a process and a technology model.

Our process model is based on the historical overload emergency service periods in Victoria, with the number of ambulances standing in line in front of the selected hospitals included according to the statistics [2]. The technology model provides the basis of telemedical solution designs, taking into account the variety of data types and the expected volume of the digital data. Mobile data coverage in various geographical areas was also considered [26], as well as the statistics of speed and general efficiency of emergency services [2].

2.1.1. The Process Model

Figure 2 describes the process of ambulatory telemedical decision making. The impact of applying the solutions built based on the technology models was assessed through probability-based simulation. This simulation used secondary data to assess the potential outcomes of our proposed solutions in the context of Victoria, Australia.

We demonstrate the potentially improved prioritisation procedure by using a discrete queuing model, where 15 selected cases arrive by ambulance to the Emergency Room (ER), in a predefined, intentionally suboptimal sequence. In this model, we align the distribution of cases with the overall approximate distribution of emergency cases delivered by ambulances to the ER in Victoria. Subsequently, we set up a continuous queuing model, working with a queue of 15 cases, applying identical probability distribution. We will demonstrate a sequential optimisation simulation in which we gradually re-prioritise medical emergencies, thus avoiding potential loss of life. We will clearly show that the use of our recommended methods could potentially save an objective set of human lives.

We justify utilising secondary data for modelling by the potentially large amount of resources required to engage in a randomised-control-trial-based approach, as is often employed in medical research. An initial assessment of the potential impact of the proposed telemedical solutions in ambulances revealed that it could improve the effectiveness of

emergency care; however, that will need to be validated with ER observations, stakeholder experiences and post hoc performance data.

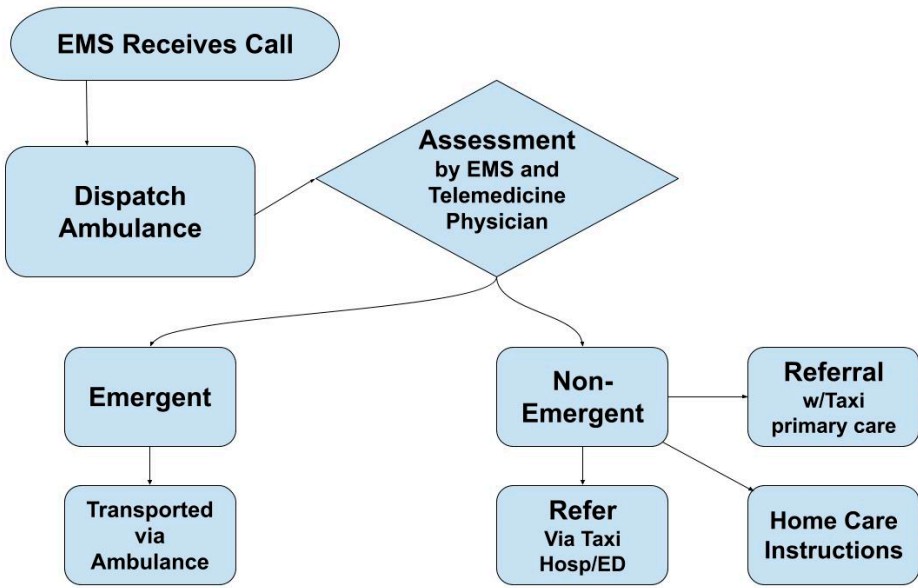

**Figure 2.** Ambulatory telemedicine.

### 2.1.2. The Technology Model

The first component of the proposed system is located in the hospital, where information is received from the site or ambulance. The counterpart to this is located on-site or in the ambulance, and serves the purpose of remote sensing and communication facilitation.

We outline the technology environment for our simulations as follows: (1) The technology environment for the emergency department workstation consists of a large monitor or 3 aligned monitors supplied to each operator. There is a software environment able to display data and provide graphical illustrations as the information is received, while recognising that even partial information can be used for assessment. (2) A sophisticated medical decision support system is applied for the main modalities.

Operators are connected in sequence and complicated cases can be discussed by rapid group meetings, by sharing each other's screens and integrating information using integrative graphical illustrations and dashboards. Table 1 provides details of the technical protocols utilised in the setup. Figure 3 illustrates a preferred embodiment of the telemedical system user interface. Table 2 provides details of the applied medical sensor technologies, subsystems and interfaces.

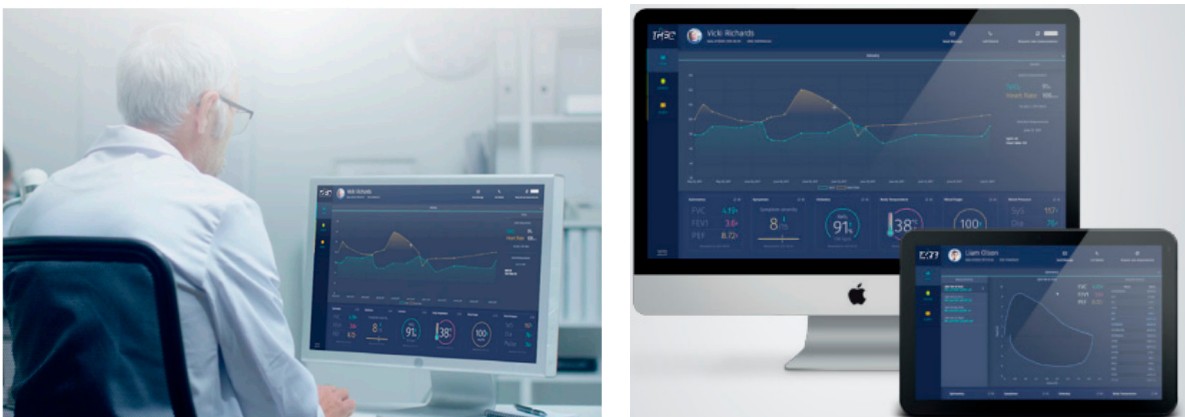

**Figure 3.** Telemedical triage station (illustration, drawn image).

**Table 1.** Medical data interfaces, protocols and formats.

| Protocol | Characteristics | Modalities |
|---|---|---|
| HL7 | Health Level 7, global med-tech data standard | ECG, respiratory, vital signs |
| XML | General data standard, widely configurable, not only medical data | Vital signs, descriptive health data |
| DICOM | Imaging specific standard | Ultrasound, optical, thermal, X-ray-based and also advanced imaging including mobile CT |
| Video feeds | VGA or any higher-resolution real-time data transfer | Vital signs, general patient observation |
| GDT | GerateDatenTrager, originally German standard, similar to HL7, descriptive | ECG, respiratory, vital signs |
| USB over TCP/IP | Virtual telemedicine: the remote medical device is emulated to be local | ECG, spirometer, vital signs |
| Sockets over TCP/IP | Suitable for any custom medical device software | Portable diagnostic systems including rapid labs |

Source: Authors' own compilation.

**Table 2.** Technology environment of the ambulance.

| Technology | | Characteristics | Means of Use |
|---|---|---|---|
| Video (Tele-video link) | | Resolution adaptive to bandwidth, best-throughput | CINE recording, optimal 30 s CINE shots, the first one triggered by a motion detector |
| Patient monitoring systems Vital signs sensors | SpO2 | Haemoglobin saturation levels, 60–99% resolution | Blood oxygenation estimation, finger clip or ear clip, feet clip for infants |
| | Temperature | Digital thermometer | Surface patch |
| | Pulse | Derived from oximeter, ECG signal or basic sensor | Heart rate detection |
| | Breathing (respiratory rate) | May be derived from ECG signal or surface sensor or mask | Basic respiratory function detection and estimations |
| | Blood pressure | Cuff-based, ABPM preferred | Cyclic automated inflation |
| | Basic ECG | Disposable or reusable electrodes or silicone vest | Cardiovascular function estimations |
| Advanced ECG | | 12-channel advanced ECG with automated diagnostic support system, disposable or reusable electrodes or silicone vest | Detailed cardiac diagnosis, recorded in high-resolution format to be sent ahead to the hospital |
| Advanced respiratory diagnostics | | Portable emergency ventilator-derived respiratory function or CPAP; portable PFT/spirometer | Lung function testing and diagnostics |
| Defibrillator | | Automated AED, portable | Drastic emergency cardiac intervention |
| Oxygen | | Cylinder or oxygen generator | Lung function support |
| Rapid intervention items | | spine boards, cervical (neck), collars, splints, bandages | Rapid manual intervention, temporary stabilisation |
| Range of drugs and intravenous fluids | | e.g., vasodilators, pain killers, antibiotics | Infection control and stabilising drugs |
| Imaging | | Ultrasound, thermal or optical imaging, or in advanced cases, X-ray-based | Unfortunately, as most high-resolution imaging modalities are radiation-based technologies, only a very limited set of choices is available for ambulances |

Source: Authors' own compilation.

An example of the use of advanced ECG in emergency medical services is a tele-cardiology project implemented in Scandinavia [25,27], where the ambulance was equipped with ECG devices capable of tele-radiology data transfer. Detailed cardiac information was sent ahead, and by the time the ambulance arrived to the hospital, the cardiologists already had a good idea about the type and severity of the cardiac issue to be addressed.

In special circumstances, especially in case of a strongly suspected or pre-diagnosed stroke, a special mobile imaging unit may be dispatched, which may include X-ray, cranial ultrasound or even a CT device. This solution was implemented in a project in Boston, Massachusetts [28].

Table 3 provides details of the medical sensor technologies, solutions and interfaces utilised in the design of the solution. In terms of communications, in our model, we calculated with a sufficient-sized buffer how to store medical data offline in case of the ambulance driving through an uncovered (i.e., rural, underground, etc.) area. The estimated data storage varies for different modalities, including basic vital signs, but in a baseline scenario, it is estimated at 1 TB.

**Table 3.** Means of telecommunication.

| Protocol | Characteristics | Field of Use |
|---|---|---|
| Short Messaging Service/MMS | Fast, inexpensive, widely available | Still has a relevance in Australia as GSM data availability is limited |
| TCP/IP | Mobile Internet | Vital signs, descriptive health data |
| UDP/IP | Short signal-based mobile data | Vital signs, emergency information |

Sources: [26,29,30].

An advanced use of scalable telecommunication services in telemedicine was implemented in South Africa [25], where the medical diagnostic/monitoring system was able to record and store data in the portable internal computer. When it found signal, then it connected to the GSM data network, sending the data to the telemedical centre with any available network speed.

In our model, we outlined a system configured to be able to send data by any available means, starting from simple SMS data transfer, as we have described in our studies and patents [25,29,30]. Even though emergencies often occur in rural areas with limited to no wireless data communications, by the time the ambulance arrives to the hospital, it will have passed areas with good coverage. The receiving station needs to be configured in a way so that it displays any incoming data, and the overall picture can be gradually revealed to the medical professionals conducting the remote triage.

### 2.2. Triage

Triage can start immediately, at the initiation of the call, when an AI can attempt to assess the nature of the emergency by analysing the tone, contents and background noises of the call (000 or 112).

In our model, we proposed a scenario where GPS coordinates are retrieved from the mobile phone used to make the emergency call. At the very least, the cell data must be retrievable, often providing sufficient location information in rural areas. Another option is to use satellite internet services, such as Starlink, in ambulances.

### 3. Results

In the following section, we provide the simulated outcome of the application of different configurations of telemedical solutions in the emergency triage and diagnostic process. We based our simulation scenarios on data provided by the Australian Institute of Health and Welfare [31]. We configured the frequency of conditions and severity levels based on data specific to Victoria (see Table 4). The specific medical conditions were

consolidated into five condition types and five urgency levels. We evaluated patient outcomes based on the time to hospital, which is essentially the time to the ER.

**Table 4.** Distribution of ambulance-driven ER admissions in Victoria.

| | | Urgency | | | | | | Estimated Number of Ambulances |
|---|---|---|---|---|---|---|---|---|
| | | Resuscitation (5) | Emergency (4) | Urgent (3) | Semi-Urgent (2) | Non-Urgent (1) | Total | |
| Condition type | Injury | 3.1% | 20.9% | 42.3% | 32.4% | 1.2% | 19.9% | 3 |
| | Infection | 1.7% | 21.2% | 50.6% | 24.6% | 2.0% | 7.2% | 1 |
| | Mental | 2.8% | 23.9% | 52.0% | 17.4% | 3.9% | 6.4% | 1 |
| | Other | 1.7% | 28.3% | 52.3% | 17.3% | 0.5% | 60.7% | 9 |
| | Cardiac | 9.7% | 53.2% | 32.2% | 4.8% | 0.1% | 5.8% | 1 |
| | TOTAL | 2.5% | 27.5% | 49.0% | 20.1% | 0.9% | 100% | 15 |

Source: [31].

Victorian ER presentations constituted 23% of all Australian presentations in 2021–2022, with slightly lower instances for the level 5 and level 1 urgency categories [31]. This is slightly lower than the share of Victoria in Australia's population (approx. 25%) [1].

*3.1. Core Vital Signs*

3.1.1. ECG Data Only

In a core scenario, we assume an obvious cardiac issue, where the ambulance staff is able to make contact with the hospital and a suitable ECG device is available. Based on an average scenario derived from general statistics [2], the following scene described in Table 5 is assumed. The distribution of cases and urgency levels is based on actual ER presentation data in Victoria (as described above), the treatment requirement timelines are adopted from the healthcare standards [31]; ambulance time to hospital and patient survival times are reasonable but arbitrary parameters.

**Table 5.** No telemedicine applied.

| Sequence Number | Case ID | Case Type | Urgency | Requires Treatment Time upon Arrival | Time to Hospital (Hours) | Est. Minimum Survival Time | Late for Medical Attention |
|---|---|---|---|---|---|---|---|
| 1 | AMB1 | Other | 3 | within 30 min | 0.50 | 6–72 h | no |
| 2 | AMB2 | Other | 2 | within 2 h | 1.00 | 24 h–7 days | no |
| 3 | AMB3 | Injury | 3 | within 30 min | 1.50 | 6–72 h | no |
| 4 | AMB4 | Other | 3 | within 30 min | 2.00 | 6–72 h | no |
| 5 | AMB5 | Mental | 3 | within 30 min | 2.50 | 6–72 h | no |
| 6 | AMB6 | Other | 3 | within 30 min | 3.00 | 6–72 h | no |
| 7 | AMB7 | Injury | 4 | withing 10 min | 3.50 | <4 h | no |
| 8 | AMB8 | Other | 4 | withing 10 min | 4.00 | <4 h | at risk |
| 9 | AMB9 | Infection | 3 | within 30 min | 4.50 | 6–72 h | no |
| 10 | AMB10 | Other | 4 | withing 10 min | 5.00 | <4 h | yes |
| 11 | AMB11 | Injury | 2 | within 2 h | 5.50 | 24 h–7 days | no |
| 12 | AMB12 | Other | 4 | withing 10 min | 6.00 | <4 h | yes |
| 13 | AMB13 | Cardiac | 5 | immediately | 6.50 | <30 min | yes |
| 14 | AMB14 | Other | 3 | within 30 min | 7.00 | 6–72 h | at risk |
| 15 | AMB15 | Other | 3 | within 30 min | 7.50 | 6–72 h | at risk |

If the staff in Ambulance 13 is equipped with advanced defibrillators and other cardiac intervention equipment, then the patient in Ambulance 13 may have a higher chance of survival. However, attending to the ambulances at the emergency department in the sequence of arrival is an extremely inefficient way of using the time of the emergency services staff.

In a more efficient scenario, the ambulances are equipped with a 12-channel ECG with automated interpretation [32] for a quick advanced diagnosis. The staff in Ambulance 13 records the ECG data right at the emergency site and transfers the information ahead to

the hospital emergency triage team. They derive a basic understanding and forward the detailed ECG data, preferably in DICOM or HL7 format [33], to the interventional cardiology department. Upon receipt, physicians in the interventional cardiology department prepare for receiving the patient and set up the advanced cardiac intervention equipment and drugs.

This will change the order of the incoming ambulances, as Ambulance 13 needs to be serviced first by the emergency department (see Table 6).

**Table 6.** Cardiac intervention support by tele-cardiology.

| Sequence Number | Case ID | Case Type | Urgency | Requires Treatment Time upon Arrival | Time to Hospital (Hours) | Est. Minimum Survival Time | Late for Medical Attention |
|---|---|---|---|---|---|---|---|
| 1 | AMB13 | Cardiac | 5 | immediately | 0.50 | <30 min | no |
| | | | | . . . | | | |
| 7 | AMB6 | Other | 3 | within 30 min | 3.50 | 6–72 h | no |
| 8 | AMB7 | Injury | 4 | within 10 min | 4.00 | <4 h | at risk |
| 9 | AMB8 | Other | 4 | within 10 min | 4.50 | <4 h | yes |
| 10 | AMB9 | Infection | 3 | within 30 min | 5.00 | 6–72 h | no |
| 11 | AMB10 | Other | 4 | within 10 min | 5.50 | <4 h | yes |
| 12 | AMB11 | Injury | 2 | within 2 h | 6.00 | 24 h–7 days | no |
| 13 | AMB12 | Other | 4 | within 10 min | 6.50 | <4 h | yes |
| 14 | AMB14 | Other | 3 | within 30 min | 7.00 | 6–72 h | at risk |
| 15 | AMB15 | Other | 3 | within 30 min | 7.50 | 6–72 h | at risk |

### 3.1.2. ECG Data and Vital Signs

In a more sophisticated scenario, we assume that the ambulance is equipped with advanced vital signs monitors, high-resolution video and auxiliary diagnostic equipment.

By performing an initial physical examination on the patient, based on signs and symptoms, the ambulance staff may be able to set up an initial diagnosis, which may further narrow down the likely status of the trauma patient. Also, this may provide a basis for distinguishing actual trauma patients from those with other mental health issues or other physical conditions, e.g., infections. High-resolution video and an appropriately formatted set of vital signs [34] sent ahead, via means of telemedicine, will enable the hospital-based triage and intervention team to further prioritise and prepare for the arrival of the patients. Hence, the order of the ambulances received shall be optimised further by prioritising cases in Ambulances 7, 8, 9 and 12, as shown in Table 7.

**Table 7.** Tele-ECG and vital sign monitoring.

| Sequence Number | Case ID | Case Type | Urgency | Requires Treatment Time upon Arrival | Time to Hospital (Hours) | Est. Minimum Survival Time | Late for Medical Attention |
|---|---|---|---|---|---|---|---|
| 1 | AMB13 | Cardiac | 5 | immediately | 0.50 | <30 min | no |
| 2 | AMB8 | Other | 4 | within 10 min | 1.00 | <4 h | no |
| 3 | AMB7 | Injury | 4 | within 10 min | 1.50 | <4 h | no |
| 4 | AMB9 | Other | 4 | within 10 min | 2.00 | <4 h | no |
| 5 | AMB12 | Other | 4 | within 10 min | 2.50 | <4 h | no |
| 6 | AMB1 | Other | 3 | within 30 min | 3.00 | 6–72 h | no |
| | | | | . . . | | | |
| 11 | AMB6 | Other | 3 | within 30 min | 5.50 | 6–72 h | no |
| 12 | AMB10 | Infection | 3 | within 30 min | 6.00 | 6–72 h | at risk |
| 13 | AMB11 | Injury | 2 | within 2 h | 6.50 | 24 h–7 days | no |
| 14 | AMB14 | Other | 3 | within 30 min | 7.00 | 6–72 h | at risk |
| 15 | AMB15 | Other | 3 | within 30 min | 7.50 | 6–72 h | at risk |

### 3.1.3. ECG Data, Vital Signs, Rapid Diagnostic Kits and Mobile Scanner

In an even more advanced scenario, the ambulance is also equipped with advanced means of rapid lab diagnostics [35]. This is one of the fastest-growing areas in medical technology; biosensors, PCR tests, liquid biopsy and other means of rapid laboratory solutions (such as CBRN monitoring) are increasingly used both in medical and related areas. In fact, a novel pathogen detected by emergency services may identify the ground zero of an infectious outbreak.

Primary infectious pathogens include bacteria, parasites, viruses, fungi and protozoa. Theoretically, all pathogens could be detected and identified by medical technology. In practice, this is not feasible due to the extreme number of pathogens normally existing on any surface or in or around any living organism.

One solution to test for the most commonly accruing and known pathogens is rapid tests. Various technologies exist for rapid testing; the most common ones include rapid reagents, PCR and chemical methods.

A more advanced means of detection is to use mobile scanners, combined with appropriate samplers, which can record a predetermined area of pathogens and nanoparticles and record a snapshot in extremely high resolution and in multiple spectra. In the era of increased occurrence of dangerous outbreaks, it would make sense to equip emergency service and first responders with such mobile scanners. If the symptoms of the patient lead the staff to suspect an infection, but the traditional rapid tests do not provide any recognisable results, the mobile scanner could take a snapshot and send the high-resolution DICOM images ahead by means of tele-pathology, to the hospital or emergency medical centre. If the local pathology experts are not able to recognise any characteristic pathogen but other conditions point to a possible outbreak, the pathogen snapshots could be forwarded via advanced telemedical channels to potential participating CBRN laboratories such as national disease centres.

If the pathogen is successfully identified or a suspected infection is characterised with high priority, the telemedical triage team may reshuffle the arriving ambulances, by prioritising Ambulance 10, as shown in Table 8.

**Table 8.** Multi-modality tele-monitoring.

| Sequence Number | Case ID | Case Type | Urgency | Requires Treatment Time upon Arrival | Time to Hospital (Hours) | Est. Minimum Survival Time | Late for Medical Attention |
|---|---|---|---|---|---|---|---|
| 1 | AMB13 | Cardiac | 5 | immediately | 0.50 | <30 min | no |
| 2 | AMB8 | Other | 4 | within 10 min | 1.00 | <4 h | no |
| 3 | AMB7 | Injury | 4 | within 10 min | 1.50 | <4 h | no |
| 4 | AMB9 | Other | 4 | within 10 min | 2.00 | <4 h | no |
| 5 | AMB12 | Other | 4 | within 10 min | 2.50 | <4 h | no |
| 6 | AMB10 | Infection | 3 | within 30 min | 3.00 | 6–72 h | no |
| 7 | AMB1 | Other | 3 | within 30 min | 3.50 | 6–72 h | no |
| | | | | . . . | | | |
| 12 | AMB6 | Other | 3 | within 30 min | 6.00 | 6–72 h | no |
| 13 | AMB11 | Injury | 2 | within 2 h | 6.50 | 24 h–7 days | no |
| 14 | AMB14 | Other | 3 | within 30 min | 7.00 | 6–72 h | at risk |
| 15 | AMB15 | Other | 3 | within 30 min | 7.50 | 6–72 h | at risk |

In a preferred scenario, the Ambulance 10 team as well as the emergency receiving department would be notified to put on proper hazmat suits and prepare for infection control by all other means possible.

### 3.2. Tele-Mental Health

The importance and necessity of tele-mental health is seriously underestimated [36]. In many cases, as statistics indicate [2], mental health issues can be serious, but at the same time patients with such issues may also block access for others to more urgent medical

interventions. Often, the only necessary or possible intervention is to talk to the patient. Since ambulance staff are not mental therapists, connecting to a professional psychologist or other mental health expert via tele-video conferencing and telemedicine can satisfy the primary needs of the patient. In many cases, the patient should not be transported to the emergency department of the hospital, but rather to a specialist institution or other alternative location.

Support by mental therapists through telemedicine can help other patients as well (in shock or trauma), and even the ambulance staff, as they are often under heavy pressure. By identifying and appropriately addressing the patient mental health issue in Ambulance 5, and applying the previous methods to optimise the ambulance priority sequence for Ambulances 14 and 15, the priority order of incoming ambulances can be improved, as shown in Table 9.

**Table 9.** Tele-mental health support.

| Sequence Number | Case ID | Case Type | Urgency | Requires Treatment Time upon Arrival | Time to Hospital (Hours) | Est. Minimum Survival Time | Late for Medical Attention |
|---|---|---|---|---|---|---|---|
| 1 | AMB13 | Cardiac | 5 | immediately | 0.50 | <30 min | no |
| 2 | AMB8 | Other | 4 | within 10 min | 1.00 | <4 h | no |
| 3 | AMB7 | Injury | 4 | within 10 min | 1.50 | <4 h | no |
| 4 | AMB9 | Other | 4 | within 10 min | 2.00 | <4 h | no |
| 5 | AMB12 | Other | 4 | within 10 min | 2.50 | <4 h | no |
| 6 | AMB10 | Infection | 3 | within 30 min | 3.00 | 6–72 h | no |
| 7 | AMB14 | Other | 3 | within 30 min | 3.50 | 6–72 h | no |
| 8 | AMB15 | Other | 3 | within 30 min | 4.00 | 6–72 h | no |
| 9 | AMB5 | Mental | 3 | within 30 min | 4.50 | 6–72 h | no |
| 10 | AMB1 | Other | 3 | within 30 min | 5.00 | 6–72 h | no |
| 11 | AMB2 | Other | 2 | within 2 h | 5.50 | 24 h–7 days | no |
| 12 | AMB3 | Injury | 3 | within 30 min | 6.00 | 6–72 h | no |
| 13 | AMB4 | Other | 3 | within 30 min | 6.50 | 6–72 h | at risk |
| 14 | AMB6 | Other | 3 | within 30 min | 7.00 | 6–72 h | at risk |
| 15 | AMB11 | Injury | 2 | within 2 h | 7.50 | 24 h–7 days | no |

### 3.3. Advanced Mobile Imaging

Mobile stroke units [28,37,38] are good examples of advanced medical imaging in transportable emergency services. The key concept is a truck or ambulance equipped with a mobile CT device, which can be driven to the patient in case of a suspected stroke. Strokes require extreme fast intervention, typically in a maximum of a few hours; otherwise, permanent brain damage may occur. If the mobile stroke unit drives towards the patient, while a family member drives in the opposite direction, they may meet in the middle and stroke intervention may be fastest and the most optimal under any circumstances. Since the mobile CT generates standard DICOM images, the data can be sent ahead via a tele-PACS system and radiologists can evaluate CT images and set up a diagnosis remotely. TPA may be administered on-site or, in case the findings allow [35], the patient may be transported to the nearest emergency care facility for rapid intervention. Figure 4 illustrates the configuration of a mobile stroke unit.

### 3.4. Continuous Queue Modelling

As demonstrated by applying the different remote and mobile diagnostic solutions to re-sequence the queue of ambulances, the potential survival of patients arriving with serious conditions demanding immediate attention can be improved. In order to approximate this improvement, we configured a continuous queue model, applying the probabilities displayed in Table 10, calculated based on data from the Australian Institute of Health and Welfare [31].

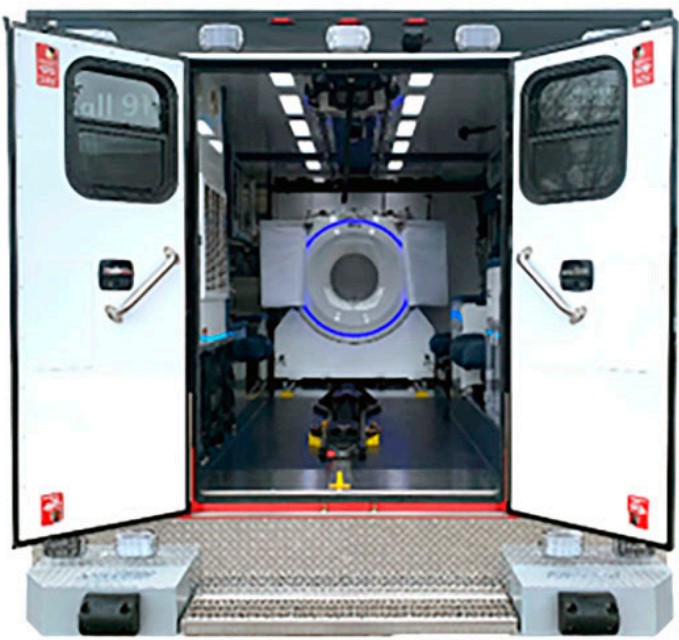

**Figure 4.** Mobile stroke unit.

**Table 10.** Recalculated distribution of ambulance-driven ER admissions in Victoria.

| | P(A) | | Resuscitation (5) | Emergency (4) | Urgent (3) | Semi-urgent (2) | Non-urgent (1) | Total |
|---|---|---|---|---|---|---|---|---|
| | | | **Urgency** | | | | | |
| Condition type | | Injury | 0.63% | 4.17% | 8.43% | 6.45% | 0.24% | 19.9% |
| | | Infection | 0.12% | 1.52% | 3.63% | 1.76% | 0.14% | 7.2% |
| | | Mental | 0.18% | 1.53% | 3.34% | 1.12% | 0.25% | 6.4% |
| | | Other | 1.00% | 17.21% | 31.74% | 10.48% | 0.29% | 60.7% |
| | | Cardiac | 0.56% | 3.07% | 1.86% | 0.27% | 0.01% | 5.8% |
| | | TOTAL | 2.50% | 27.50% | 49.00% | 20.10% | 0.90% | 100.00% |
| | P(A$_{15}$) | | Resuscitation (5) | Emergency (4) | Urgent (3) | Semi-urgent (2) | Non-urgent (1) | |
| Condition type | | Injury | 0.04% | 0.27% | 0.54% | 0.42% | 0.02% | |
| | | Infection | 0.01% | 0.10% | 0.24% | 0.12% | 0.01% | |
| | | Mental | 0.01% | 0.10% | 0.22% | 0.07% | 0.02% | |
| | | Other | 0.07% | 1.07% | 1.86% | 0.67% | 0.02% | |
| | | Cardiac | 0.04% | 0.20% | 0.12% | 0.02% | 0.00% | |

Source: [31]. Information is rounded at 2 decimal accuracy. Please contact authors for exact figures.

Firstly, using the Bayes theorem of conditional probability, we recalculated Table 4, to show the distribution of cases at various severity levels.

$$P(A|B) = \frac{P(B|A)P(A)}{P(B)} \tag{1}$$

P(A|B) represents the distribution of different case groups provided that they belong to a particular condition type (see Table 4). P(B) is the distribution of all cases across different condition types. P(A) is the distribution of all cases across different condition types and severity levels. And finally, P(B|A) = 1 represents the probability of a condition type being assigned an injury level.

$$P(A) = \frac{P(A|B)P(B)}{P(B|A)} = P(A|B) * P(B) \tag{2}$$

Based on Equation (2), the distribution of all cases of different condition types and severity levels can be calculated (see Table 10).

We model a queue of 15 ambulances. We know the probability of the occurrence of a case (P(A)) across all possible cases, that is, for the whole queue. In order to calculate the probability of a case occurring in a particular position in the queue, we need to recalculate the distribution of probabilities (see Formula (3)), utilising the concept of complementary probability.

$$P(A) = P(A_{15}) + 1)^{15} - 1 \tag{3}$$

Based on Formula (3), $P(A_{15}) = 1 - (1 - P(A))^{\left(\frac{1}{15}\right)}$ can be calculated, as displayed in Table 10.

In order to approximate the potential improvement with the introduction of remote diagnostic and telemedical solutions, we need to calculate the probability of cases occurring later in the queue potentially not receiving timely medical attention. Using the concept of complementary probability, we can calculate the probability of a particular case occurring not in the first place of the 15-case-long queue using Formula (4)

$$P\left(A \text{ not in the } 1^{\text{st}} \text{ in queue}\right) = (1 - P(A_{15})) * \left(1 - (1 - P(A_{15}))^{14}\right) \tag{4}$$

Similarly, we can calculate the probability of a particular case or cases occurring not in the first or second place of the queue using Formula (5)

$$P\left(A \text{ not in the } 1^{\text{st}} \text{ or } 2^{\text{nd}} \text{ in queue}\right) = (1 - P(A_{15}))^2 * \left(1 - (1 - P(A_{15}))^{13}\right) \tag{5}$$

Table 11 shows the probabilities of certain case groups occurring later in the ambulance queue, thus presenting a risk of lack of timely attention and treatment. The calculations show that approximately 2.3% of very urgent (urgency level 5) cases occur not in the first place of the queue. Approximately 25.1% of serious cases (urgency level 4 or 5) fall later than the first place in the queue. Approximately 51.8% of urgency level 3, 4 or 5 cases occur later than the second place in the queue, suggesting a potential risk imposed by the lack of proper ambulance sequencing.

**Table 11.** Risk of late ER admissions across certain condition types and severity levels in Victoria.

| | **Probability of Case Occurring** | | | | |
| --- | --- | --- | --- | --- | --- |
| | **Overall** | **In any One Step of a Sequence of 15** | **Not in the First Step of a Sequence of 15** | | |
| Cardiac—level 5 | 0.56% | 0.04% | 0.52% | | |
| Cardiac—levels 4/5 | 3.63% | 0.24% | | 3.27% | |
| Cardiac—levels 3/4/5 | 5.48% | 0.36% | | | 4.50% |
| Injury—level 5 | 0.63% | 0.04% | 0.58% | | |
| Injury—levels 4/5 | 4.80% | 0.31% | | 4.28% | |
| Injury—levels 3/4/5 | 13.23% | 0.83% | | | 10.12% |
| Infection—level 5 | 0.12% | 0.01% | 0.11% | | |
| Infection—levels 4/5 | 1.64% | 0.11% | | 1.51% | |
| Infection—levels 3/4/5 | 5.27% | 0.34% | | | 4.34% |
| Mental—level 5 | 0.18% | 0.01% | 0.17% | | |
| Mental—levels 4/5 | 1.71% | 0.11% | | 1.57% | |
| Mental—levels 3/4/5 | 5.05% | 0.33% | | | 4.17% |
| Other—level 5 | 1.00% | 0.07% | 0.93% | | |
| Other—levels 4/5 | 18.22% | 1.12% | | 14.45% | |
| Other—levels 3/4/5 | 49.96% | 2.74% | | | 28.66% |
| Total | - | - | 2.31% | 25.1% | 51.8% |

## 4. Discussion

We have observed bottlenecks and inefficiencies in emergency care in our region. We have modelled and investigated a range of telemedical means [24,39–48] and methods to support and optimise the current ambulance system, with the understanding that all proposed solutions for any healthcare service are subject to trade-offs, and, therefore, perfect solutions are not possible to offer.

We built our technology models based on examining, building and testing a range of telemedical applications [24,25,42,47,49–52]. Our models were based on a significant body of prior experience and were optimised to the environment in Victoria, Australia based on the actual experience of the researchers, and with the knowledge of chronic inefficiencies of the system [4,5,9,10].

### 4.1. Barriers to Implementing Telemedical Solutions in Emergency Care

While the benefits of using telemedicine in emergency care are relatively easy to model and demonstrate, there are a number of obstacles and barriers in implementation [53]. Several of these obstacles have been described in earlier publications [24,25]. While usability, education [54], technical service and infrastructure availability are significant barriers, cost remains the most significant issue, closely followed by the associated bureaucracy.

### 4.1.1. Protocols and Bureaucracy

Medical device and technology bureaucratic protocols are on a sharp rise with the introduction of the EU MDR. This is expected to increase costs globally, and compounded by other economic effects and supply chain challenges currently experienced in the global economy, a cost barrier to telemedical technology investments emerges. Further research is needed to investigate the costs and efficiency of emergency telemedicine.

### 4.1.2. Data Privacy and Security

Medical data are equally as sensitive as financial or government-owned information and similar methods need to be implemented to protect them [55]. Most global medical interface standards, including HL7 and DICOM, contain protocols for medical data encryption and transfer. In addition to standardised interfaces and protocols, custom local implementations are usually required. Anonymous data transfer is needed in most cases, if strong encryption is not available or is impractical. However, the balance between the level of security and ease of use presents an unavoidable trade-off in design [41]. Further research can explore these trade-offs in terms of design and implementation of emergency telemedical solutions.

### 4.1.3. Difficulties of Economic Assessment

Telemedicine is regarded not only as clinically effective but also economically beneficial. Davalos et al. [56] highlight the significant gaps in the economic assessment of telemedicine. Issues include limitations of generalisability, methodological inconsistency and standardisation, and relatively poor-quality and short-term data. Although Moffatt and Eley [57] highlight that telemedicine has been perceived as useful in a rural Australian context, costs and resourcing have been significant barriers to implementation. Further analysis provided by Mehrotra et al. [58] during and after the pandemic highlighted the complex practical and policy issues in relation to funding telemedical activities in Australia and globally. However, methods appropriate for the economic assessment of telemedical solutions for emergency care are not well developed, and the moral and ethical issues inherent in emergency care are difficult to align with economic rationale [59].

In light of the above scholarly views and moral dilemmas, in our article, we recognise the importance of the economic assessment of telemedical solutions, but do not extend our model and analysis to incorporate this complex issue. Further research will be needed to explore the economic aspects of the proposed telemedical solutions, as well as their benefits, with specific focus on defining clear patient outcomes and outcomes for other stakeholders.

### 4.2. Potential Impact of the Proposed Solution

When applying the proposed remote and telemedical solutions, the most urgent cases (approximately 2.31% of all cases, as shown in Table 11) can be re-prioritised, avoiding potential loss of life. Among all case groups, cases labelled as 'other' are the most frequently occurring, followed by injuries and cardiac cases. In Victoria alone, the total estimated number of emergency room presentations by ambulance annually was 2,262,000 in the financial year 2021–2022 [31]. This means that the proposed system upgrade would have resulted in improved outcomes for approximately 52,000 cases annually in Victoria.

### 4.3. Machine Learning in Telemedicine

We mentioned the potential use of AI in our models, assisting the operators with responding to emergency situations. We see potential in the application of machine learning [60] and deep learning [61]-based solutions, whereby telemedical operators can be assisted in terms of making better diagnostics, interpreting the incoming information faster and more effectively, and building validated knowledge bases to assist pre-arrival diagnostic work. Furthermore, AI-type solutions can help bridge the gap between intervals of data interruptions, and potentially eliminate the impact of corrupted or missing information on the processes.

### 4.4. Model Estimate Uncertainties

Our model uses Bayes' theorem to estimate the conditional probability of certain occurrences (such as the first ambulance not being the most urgent one). The deterministic nature of this theoretical foundation [62] implies that the modelling results will be sensitive to inputs and assumptions. Model inputs include the distribution of ER presentations across urgency categories by ambulance, which is estimated from ER presentation and ambulance usage data. A specific and more accurate dataset, which gives data on actual ER presentations by ambulance in different urgency categories and symptom types, could reduce uncertainties in the model outcomes.

Some parameters of the model were fixed estimates. Ambulance time to hospital and patient survival time were determined as arbitrary input parameters. Changing these will alter the model outcomes and the conclusions. A sensitivity analysis of the model to these parameters can be conducted in the future to evaluate the extent to which the parameter changes influence model results.

### 5. Conclusions, Recommendations and Further Research

Our model demonstrates an improvement potentially measurable in human lives, which is impossible to quantify in financial terms. Furthermore, it clearly shows potential improvements in quality of service if the telemedical methods are applied.

We strongly suggest the adoption and implementation of tele-cardiology, telemedicine, tele-mental health and, in justified cases, tele-radiology in emergency services. We estimate a potential of over 52,000 lives impacted annually in Victoria alone.

Limitations to our model include assumptions of the distribution of data, and the length of the ambulance queues used in the continuous queue model. Financial, investment considerations, cost–efficiency ratio, patient data privacy and other matters are still to be investigated in detail. Adjusting these parameters may change the outcomes of the modelling, and we propose a sensitivity analysis of these parameters to be conducted for future research.

In the future, we intend to conduct in-field studies in cooperation with local hospitals and healthcare centres to test the effectiveness and further optimise the discussed methods through testing and practical application. Further research will need to focus on validating the designs presented, and their impact, through a sequence of pilot studies, experiments and clinical trials. In terms of the cost–benefit analysis, once the designs are validated, further analysis is needed to assess costs and benefits associated with these telemedical solutions. Finally, considerations need to be given regarding the accessibility of these

solutions, particularly in terms of the readiness of both staff and patients to adopt the new technologies, and the implications for vulnerable and disadvantaged groups. Following this idea, as the implementation of such systems and solutions is a complex endeavour, further research needs to explore the training needs and application issues reported by professionals in the sector. Alternatively, a specification of patient outcomes, and, in particular, clinical insights from the use of telemedical solutions, can also be explored.

**Author Contributions:** Conceptualisation: G.L.F. and Á.P.; methodology: G.L.F. and Á.P.; software: G.L.F.; formal analysis: Á.P.; investigation: G.L.F. and Á.P.; writing: G.L.F. and Á.P. All authors have read and agreed to the published version of the manuscript.

**Funding:** This research received no external funding.

**Data Availability Statement:** Not applicable.

**Conflicts of Interest:** The authors declare no conflict of interest.

**Abbreviations**

| | |
|---|---|
| AI | Artificial Intelligence |
| CAD | Computer-Aided Dispatch |
| CBRN | Chemical, Biological, Radiological and Nuclear |
| CT | Computer Tomography |
| DICOM | Digital Imaging and Communications in Medicine |
| ECG | Electrocardiography |
| ER | Emergency Room |
| ESTA | Emergency Services Telecommunication Authority |
| GPS | Global Positioning System |
| GSM | Global System for Mobile Communications |
| HL7 | Health Level Seven |
| IT | Information Technology |
| MMS | Multimedia Messaging Service |
| NSW | New South Wales |
| PACS | Picture Archiving and Communication System |
| PCR | Polymerase Chain Reaction |
| SMS | Short Message Service |
| TB | Terabyte |
| TCP/IP | Transmission Control Protocol/Internet Protocol |
| tPA | Tissue Plasminogen Activator |
| EU MDR | European Union Medical Device Regulation |

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
