# Peer review of "Modelling the Application of Telemedicine in Emergency Care"

_inventions, doi:10.3390/inventions8050115_

Round 1

Reviewer 1 Report

Estimated authors,

first of all, thank you for this very interesting paper, whose content is both innovative and highly sound from a scientific point of view.

In this conceptual study, Ferenczi and Perényi suggest that a radical implementation of telemedicine could lead to substantial sparing in terms of human life, particularly for certain clinical conditions requiring emergency care.

Authors estimate in around 50,000 the number of lifes impacted by years. This is a very huge number, and could represent both the main strength and the major flaw of this study. On the one hand, the message that is conveyed through this study is clear: telemedicine could be used even in critical care (see subsection on stroke units) and its usage will save life. On the other hand, how Authors estimate in 52,000 the impacted events remains unclear across the main section, and some amendments of the main text in order to transparently provide this estimate could be quite important.

Another issue that Authors should discuss is from the economic point of view. Usually telemedicine is regarded not only as clinically effective, but also as instrumental for saving money and resources (that in turn lead to economic sparing): have you provided any calculation about the costs and sparing associated with your program?

Moreover, Authors should make some minor improvements:

- please provide the meaning of all shortenings and acronyms the first time you provide them across the main text, even across the main labels of tables and figures;

- please uniform the number of decimals in the figures of tables;

- Table 1 and 2 could be moved as annex materials, the same for figure 1, 3 and 4, and the subfigure on the right side of figure 2.

- please evaluate whether any of Tables 5 to 9 could be merged.

Author Response

Dear Reviewer,

Thank you for providing your feedback and suggestions. We attempted to improve the manuscript, and respond to your questions. We hope that the revised version satisfies your concerns, and look forward to an opportunity to address any further recommendations you may have.

Please find a detailed report to your feedback in the uploaded file.

Regards,

The authors

Reviewer 2 Report

The study aims to explore the potential benefits of telemedicine in improving the efficiency of ambulance services in Victoria, Australia. Here are a few critical concerns that should be considered:

1. The study primarily uses statistical analysis and theoretical models to explore the potential benefits of telemedicine. While this approach can generate useful insights, it would be more persuasive if supported by experimental data from pilot studies or trials showing the impact of telemedicine in this context.

2. The use of telemedical technologies assumes that patients have access to the necessary hardware and internet connectivity and can use the technology effectively. The study should address potential barriers to technology accessibility and readiness, particularly for disadvantaged or vulnerable groups.

3. Telemedical technologies involve the transmission of sensitive health data, which raises concerns about data privacy and security. The study should discuss how these concerns would be addressed.

4. Implementing telemedical technologies involves integrating them with existing healthcare systems, which can be complex and challenging. The study should discuss how this would be achieved, and what potential barriers might exist.

5. The successful implementation of telemedicine depends not only on patient acceptance, but also on healthcare staff being willing and able to use the technology effectively. The study should address the need for staff training and strategies for encouraging staff acceptance.

6. The study mentions the use of "novel telemedical technologies", but does not specify what these technologies are. More information on the specific technologies being proposed would be helpful in evaluating the feasibility and potential effectiveness of this approach.

7. More references on different modeling in healthcare should be added to attract a broader readership i.e., PMID: 37112302, PMID: 36174933.

8. While the study focuses on ambulance services in Victoria, Australia, it would be helpful to compare these findings with similar services in other regions, both within Australia and internationally. This would provide context and help to determine whether the findings are applicable elsewhere.

9. The study should provide a cost-effectiveness analysis comparing the proposed telemedical technologies with the current standard of care. This is important for policy makers and healthcare providers when considering implementing these technologies.

10. It is crucial that any new system or technology implemented in healthcare improves patient outcomes. The study should clearly define and measure these clinical outcomes to evaluate the effectiveness of the proposed telemedical technologies.

11. Uncertainties of models should be reported.

12. Quality of figures should be improved.

13. More discussions should be added, especially regarding clinical insights of models.

14. It is like the authors included a lot of references from internet news. It is better to replace them with some scientific articles.

15. Overall, English writing and presentation style should be improved.

Overall, English writing and presentation style should be improved.

Author Response

(The authors gave the same response as above.)

Reviewer 3 Report

This paper describes a method of telemedicine to handle the bottleneck of patients/procedures found in the emergency ward of hospital. The system/method of telemedicine minitores the patients signs, and interventions/procedures.

I am not sure if the telemedicine triage station is located on the hospital or in the ambulance, and if the monitoring of signs of the patients is done in the hospital or in the ambulance as well (maybe both).

Additional comments:

(1) It is mentioned that the ambulances usually wait in line in the entrance of the hospital. Does this leave patients waiting for ambulances at their homes? Cannot the hospital increase the number of doctors, nurses, and beds at emergency ward?

(2) Table 1 shows the medical data interfaces, protocols, and format. Looking at the content, some looks like a medicalized ambulance. Could you please explain if all the protocols are to be included in the emergency room or also some in the ambulance? Mobile CT is impressive but I am not sure if the other equipment would fit in the ambulance and/or if it would be more efficient to drive to the nearest hospital.

(3) Is the telemedical triage station connected to the ambulance and all the signs monitored as seen in Figure 3?

(4) Do the telemedical triage station uses any type of AI algorithm to evaluate the patients disease, severity, an prioritation for treatment?

(5) Are all the results are based on a simulation? If so, this invention has not been executed yet?

(6) Line 396, "Our model demonstrates an improvement potentially measureable in human lives, which is impossible to quantify in financial terms, yet it clearly shows potential improvements in quality of service if the tele-medical methods are applied. " How was this demonstrated? Where in the results section it is shown?

(7) The discussion is short. The similarities with other systems could be discussed.

Author Response

(The authors gave the same response as above.)

Round 2

Reviewer 1 Report

The paper has been amended according to my previous recommendations. Therefore, I'm advocating its acceptance.

Good job.

The overall quality of the English text is appropriate for this journal.

Reviewer 2 Report

My previous comments have been addressed.